# The Interplay of Oxytocin and Attachment in Schizophrenic Patients: An fMRI Study

**DOI:** 10.3390/brainsci13081125

**Published:** 2023-07-25

**Authors:** Kristina Hennig-Fast, Dominik Meissner, Carolin Steuwe, Sandra Dehning, Janusch Blautzik, Dirk W. Eilert, Peter Zill, Norbert Müller, Thomas Meindl, Maximilian Reiser, Hans-Jürgen Möller, Peter Falkai, Martin Driessen, Anna Buchheim

**Affiliations:** 1Department of Psychiatry and Psychotherapy, Ludwig-Maximilians University, 80336 Munich, Germanyhans-juergen.moeller@med.uni-muenchen.de (H.-J.M.); peter.falkai@med.uni-muenchen.de (P.F.); 2Department of Psychiatry and Psychotherapy, University of Bielefeld, 33615 Bielefeld, Germany; 3Department of Radiology, Ludwig-Maximilians University, 81377 Munich, Germany; 4Department of Psychology, University Innsbruck, 6020 Innsbruck, Austria

**Keywords:** attachment, interpersonal functioning, theory of mind, social cognition, social emotion, oxytocin, schizophrenia, neuroimaging, fMRI

## Abstract

Background: Attachment theory offers an important framework for understanding interpersonal interaction experiences. In the present study, we examined the neural correlates of attachment patterns and oxytocin in schizophrenic patients (SZP) compared to healthy controls (HC) using fMRI. We assumed that male SZP shows a higher proportion of insecure attachment and an altered level of oxytocin compared to HC. On a neural level, we hypothesized that SZP shows increased neural activation in memory and self-related brain regions during the activation of the attachment system compared to HC. Methods: We used an event-related design for the fMRI study based on stimuli that were derived from the Adult Attachment Projective Picture System to examine attachment representations and their neural and hormonal correlates in 20 male schizophrenic patients compared to 20 male healthy controls. Results: A higher proportion of insecure attachment in schizophrenic patients compared to HC could be confirmed. In line with our hypothesis, Oxytocin (OXT) levels in SZP were significantly lower than in HC. We found increasing brain activations in SZP when confronted with personal relevant sentences before attachment relevant pictures in the precuneus, TPJ, insula, and frontal areas compared to HC. Moreover, we found positive correlations between OXT and bilateral dlPFC, precuneus, and left ACC in SZP only. Conclusion: Despite the small sample sizes, the patients’ response might be considered as a mode of dysregulation when confronted with this kind of personalized attachment-related material. In the patient group, we found positive correlations between OXT and three brain areas (bilateral dlPFC, precuneus, left ACC) and may conclude that OXT might modulate within this neural network in SZP.

## 1. Introduction

Humans are highly social animals, and the ability to remember, imagine, share, and predict feelings, emotions, and attitudes of selves and of other individuals is encouraged by society. Social species’ brains do not exist in isolation. Relational neuroscience, a broad term, draws on a range of experiential, neuroendocrine, and functional magnetic resonance imaging evidence to illuminate how early key relationships affect the development of brain architecture and functioning [1]. Attachment theory [2] demonstrates the role of childhood experience in shaping adult life and the importance of security for affective flexibility. Developmental and attachment research offers an important framework for interpersonal interaction experiences that play a critical role in regulating affect, cognition, and interpersonal behavior and are also related to interpersonal functioning, resilience, and mental symptoms [3,4,5,6]. According to the attachment theory, interactions with early main caregivers are memorized and organized as schematic, or script-like internal working models [7,8], the developmental base of an individual’s attachment representations. Attachment patterns reflect individual differences in beliefs about self and others, interpersonal functioning, and affect regulation, and are associated with specific neural activation patterns in various studies [9,10].

One of the most prominent psychiatric diseases, schizophrenia, is associated with severe cognitive and social impairment, leading to a loss of autonomy, professional and personal achievements, and insecure attachment relationships [11,12]. Accordingly, social functioning has been identified as one of the most important outcome variables in schizophrenia [13,14]. There is also evidence to suggest that social functioning in schizophrenia is linked with patients’ cognitive capacities [15]. Individuals who suffer from schizophrenia often show a stable impairment in many aspects of social cognition, such as emotion recognition, social perception, attachment, and empathy [11,16,17,18,19,20]. These poor abilities are consistent over time, apparent in the prodrome of psychosis [21,22,23,24], and seriously impede social competence [25,26]. One possible way to access problems associated with impaired functioning and poor outcome refers to studies that have examined concepts like social cognition, metacognition, and attachment styles in patients with schizophrenia, e.g., [27].

With regard to attachment and attachment representations, several studies demonstrated their central roles in the development of several psychiatric disorders, (see reviews [28,29]). In particular, they show that disruption in early attachment relationships by experiences of loss or trauma may influence later pathology by leading to a series of disturbed mental representations that are elaborated and consolidated over the life span.

Although attachment theory has had a significant impact on theories and research concerning the nature of human relationships [2], there is limited research investigating its relevance to psychosis and schizophrenia using the established Adult Attachment Interview (AAI) [30,31,32]. In general, study results using different attachment measures like self-reports are pretty heterogeneous.

Dozier et al. [28] reported that schizophrenic patients had more insecure-dismissing attachment representations using the AAI compared to patients with an affective disorder. This finding was replicated by Tyrrell et al. [33], who showed that 89% of schizophrenic patients were classified as insecure-dismissing. When including the “unresolved” category (four-way analysis), 44% of the patients were classified as unresolved/disorganized. Reviews on psychotic phenomenology and attachment by Sood et al. [34] and Korver-Nieberg et al. [35] summarized that insecure individuals were more vulnerable to developing maladaptive coping strategies in recovering from psychosis. Gumley et al. [36] reported in their systematic review on attachment and schizophrenic patients using self-report measures small to moderate associations between greater attachment insecurity—reflected in an anxious and avoidant attachment—and poorer engagement with services, more interpersonal problems, more avoidant coping strategies, more negative appraisals of parenting experiences and more severe trauma. The authors also found small to modest associations between attachment insecurity and more positive and negative symptoms as well as greater affective symptom problems. Moreover, Sood et al. [34] assumed that insecure attachment is likely to lead to paranoia via negative beliefs about self and others, cognitive fusion, and the use of maladaptive emotion regulation strategies. Korver-Nieberg et al. [35] considered that understanding the role of attachment in symptoms may help to gain insight into the development or persistence of symptoms. The importance of attachment experiences for processing social information, metacognitive skills, and developing social relationships, including therapeutic relationships, in samples with psychosis should be highlighted in future studies.

Attachment theory more generally has been shown to be important in understanding psychosis [37], with studies showing associations between attachment avoidance and voice-hearing (e.g., [4,38]) and paranoia [3]. Lavin et al. [12] found in a systematic review a correspondence between paranoid symptoms and an anxious-preoccupied attachment style. In the largest study to date examining attachment profiles in psychosis, Bucci et al. [39] found that unresolved/disorganized attachment was associated with a higher proportion of sexual and physical abuse and more positive symptoms, such as delusions and hallucinations, compared with other attachment patterns, suggesting disorganized attachment might be a more putative attachment pattern compared with other types of attachment for positive psychotic symptoms. In sum, all studies on schizophrenia including paranoid symptoms and attachment reported a high amount of insecure attachment however with no clear association to a specific attachment pattern.

### 1.1. Neurobiology of Attachment and Mentalizing in Psychosis: A Developmental Perspective

Debbané et al. [29] identified at least five neurobiological pathways linking attachment to risk for developing psychosis when trying to better understand the neurobiological mechanisms through which insecure attachment may contribute to psychosis. Their review considered neuroscientific and behavioral studies that underpin mentalization as a suite of processes potentially moderating the risk to transition to psychotic disorders. They developed a model where embodied mentalization would lie at the core of a protective, resilient response mitigating the adverse and potentially pathological influence of the neurodevelopmental cascade of risk for psychosis. Their review proposes an integrative model of psychosis based on three key assumptions: (1) attachment security constitutes a non-specific protective factor in individuals at increased risk for psychosis; (2) disturbed attachment can impact at least five different neurobiological pathways implicated in sustaining self and other mentalizing; and (3) embodied mentalizing may serve as a moderating factor in the expression of psychosis. Debbané et al. [29] assumed at the interpersonal level, attachment security is associated with help-seeking behavior and with more favorable outcomes in individuals suffering from psychosis. At the psychological level, attachment security provides a key developmental context to acquire the building blocks for robust social cognition and mentalizing. In some individuals, developmental adversity may promote the use of anxious, avoidant, or even disorganized attachment strategies. While these strategies may constitute adaptation attempts to adverse and hostile environments, they tend to undermine the development of the capacity to attend to one and others’ minds (mentalizing), and they affect the unfolding of social cognitive skills. The process of mentalizing may help the individual compensate for endophenotypic impairments in the integration of sensory and metacognitive information.

Imaging studies have also revealed significant abnormalities regarding mentalization in schizophrenia. In addition to the brain volumetric abnormalities in pre-frontal and temporal areas [40,41,42], associated with deficient mentalization, studies using different functional imaging procedures have undoubtedly described atypical neural activation characterized by over- and under-activation in mentalizing regions [43]. According to a meta-analysis, the mPFC, the left orbitofrontal cortex (OFC), and a small portion of the left posterior TPJ are regularly found under-activated, while over-activation was reported in the more dorsal part of the TPJ bilaterally, in the medial occipito-parietal cortex, right premotor areas, left cingulate gyrus, and lingual gyrus [44]. Moreover, different activation has been shown in high-risk patients in the right TPJ, right middle temporal gyrus (MTG), and left precuneus [45], and also in clinically asymptomatic relatives in dorsolateral PFC, dorsomedial PFC, and right inferior frontal gyrus [43,46]. Despite the extensive research on mentalizing in schizophrenia, the majority of studies have been focused only on explicit mentalizing. Relatively little is known about potential alterations of implicit mentalizing. Based on the neurodevelopmental hypothesis of schizophrenia [47], implicit mentalizing is assumed to be also impaired, as early neurodevelopmental abnormalities may affect the neural networks responsible for implicit mentalizing, which in turn may influence the development of later explicit mentalizing skills. It is suggested that the impaired early embryonic and later adolescent maturation of the PFC is likely to play a role not just in the development of behavioral, but also in cognitive symptoms of SZP [48]. Studies on childhood-onset schizophrenia emphasize the loss in parietofrontal and parietotemporal areas as well [49]. The abnormal growth process of the cingulo-fronto-temporal module development and disturbance of maturational trajectories were also reported, which affect several neuronal networks repeatedly found impaired in mentalizing studies (right IFG, triangular and opercular part; right medial orbital superior frontal gyrus; right gyrus rectus; left posterior cingulate gyrus) [50,51,52,53].

Furthermore, from the neurodevelopmental perspective, there is a well-documented influence of attachment insecurity on the hypothalamic-pituitary-adrenal (HPA) axis [54,55,56,57] and regions with a high density of glucocorticoid receptors (e.g., hippocampus and prefrontal cortex) as well as brain regions sensitive to repeated neuronal excitation (e.g., amygdala) [58,59]. Functional and structural changes in these areas have an important influence on the disruption of cognitive processes associated with psychosis. Chronic exposure to stress can produce extensive structural alterations in the PFC, including the loss of dendritic length, branching, and spine density [60].

Insecure attachment patterns may also contribute to neurodevelopmental risk through the dopaminergic and oxytocinergic systems, as well as bear influence on neuroinflammation and oxidative stress responses [61]. The prosocial effects of oxytocin have been specifically attributed to its stress-regulating properties [62], also in schizophrenia [63,64,65]. Oxytocin (OXT) has repeatedly been shown to improve different facets of social cognition, such as emotion recognition [66,67,68,69,70], empathy [71,72,73], and trust [74,75].

In schizophrenia, OXT was reported to improve emotion recognition among schizophrenic patients [76], as well as theory of mind and social judgments [77]. Another study found significant correlations between OXT and social cognitive bias in the control group and in patients with delusions, but not in patients without delusions. Social cognitive capacity only correlated significantly with OXT in patients with delusions in schizophrenic patients [78].

In schizophrenia, further studies have shown associations between reduced serum levels of oxytocin and difficulties in facial emotion identification [79], reduced plasma levels of oxytocin among patients during trust-dependent interactions with others [80], OXT plasma levels associated with severe life events and fewer important attached persons as well as related to negative symptoms of schizophrenia [81], as well as negative associations between oxytocin levels and severity of psychotic psychopathology [63]. According to Tas et al. [62], the oxytocinergic system affects social cognition by acting upon subcortical structures (i.e., amygdala) responsible for basic social cognitive processes (i.e., facial emotion recognition), which in turn affect cortical areas (i.e., PFC), responsible for higher order metacognitive processes such as theory of mind (ToM). While interpersonal arousal in the context of insecure attachment may lead to transient disruptions in cortical brain areas responsible for metacognitive processing (i.e., mPFC, posterior cingulate cortex (PCC)), in healthy subjects the oxytocinergic system modulates amygdala-based stress and promotes mutually beneficial contingent relational responses [82]. The prosocial effects of OXT appear to be particularly pertinent through their modulation of avoidant attachment responses. In the context of a genetic diathesis for psychosis, attachment insecurity and concomitant depletion of available OXT could, therefore, amplify the vulnerability to dopamine dysregulation and heighten the vulnerability to aberrant salience and low-grade psychotic symptoms. Several studies have linked variations in the OXTR gene expression to schizophrenia [83,84,85]. The findings from humans and animals suggest that early life stress may promote oxidative damage in the brain, which is shown to both alter processes of neurodevelopment [86] and enhance processes of neurodegeneration [87] that lead to a loss of neural connectivity in various brain regions, including the PFC [88]. Brent et al. [61] consistently showed in resting-state fMRI studies that greater alterations of mPFC functional connectivity among individuals at genetic high risk are associated with greater levels of psychotic manifestations.

### 1.2. Hypotheses

In our study, we examine for the first time the neural correlates of attachment and OXT in schizophrenic patients (SZP) compared to healthy controls (HC) using fMRI.

Our first hypothesis is that SZP will show a higher proportion of insecure attachment patterns compared to HC.

Our second assumption is that SZP will show altered plasma OXT levels compared to HC (see [29,81,89]).

On a neural level, we assume that SZP show during stimulation of the attachment system a significantly increased neural activation compared to HC in areas related to vivid and affective autobiographical memory (amygdala, hippocampus, precuneus, PCC, TPJ, and STS) on affect-related but also neutral stimuli.

We are also interested in the interplay of OXT and brain response. Here we do not have any directed assumptions so far.

## 2. Materials and Methods

### 2.1. Sample Characteristics

Twenty male patients with schizophrenia participated in the study matched for mean age, years of education, verbal IQ, and handedness with healthy male participants (see Table 1). All included patients were recruited from the Psychiatric Department of the Ludwig-Maximilians-University Hospital, Munich. They fulfilled the DSM-IV [90] criteria for chronic schizophrenia (paranoid, N = 13; disorganized, N = 5; undifferentiated, N = 2) as diagnosed by consensus of the current treating psychiatrists and psychologists and verified by the Structural Clinical Interview for DSM-IV (SCID-I German version, [91]), internal consistency, α ≥ 0.70). The mean duration of illness was 3.96 years (SD = 3.18). Symptom ratings obtained 2 to 4 weeks prior to scanning by the Positive and Negative Syndrome Scale (PANSS, [92]), internal consistency, α ≥ 0.70 ≤ 0.85) indicated that psychotic symptom levels were in the mild to moderate range (mean PANSS-total-score = 55.9 (13.6)) according to Leucht et al. [93]—a level that was supposed to not interfere with the ability to perform the task. There were no between-group differences in PANSS scores assessed by a Kruskal-Wallis-Test or duration of illness assessed by an ANOVA with respect to different types of diagnosis (see Table 2). Patients with current diagnoses or histories of traumatic brain injury, epilepsy, neurologic conditions, or other severe psychiatric conditions (including major depression, bipolar disorder, PTSD, and borderline personality disorder) were excluded from the study, as were patients treated with benzodiazepines. Nineteen patients had been taking atypical neuroleptics (mean dose: 500 ± 200 mg of Clozapine, Risperidone, or equivalent). An additional five patients had been receiving typical long-term neuroleptics administered at the standard dose (mean dose: 7 ± 3 mg of Haloperidol or equivalent), further, five patients were also treated by antidepressive medication (mean dose: 20 mg ± 10 mg of Citalopram, or equivalent). One patient was free of medication. Twenty healthy male participants were recruited from the community. For inclusion, they had neither current symptoms nor a history of alcoholism, drug abuse, neurological or psychiatric illness. They were supposed to be free of medication.

The healthy participants had been recruited via a newspaper advertisement. Data collection took place in parallel for patients and healthy control subjects. Study participants (SZP and HCP) were recruited verbally or by phone and were informed in advance about the contents, inclusion and exclusion criteria of the study, as well as risks associated with participation and remuneration (€50.-). Only participants who agreed a priori to participate in the study were recruited.

The healthy comparison group was a sample parallelized with respect to age, gender, and schooling. Additional exclusion criteria concerning the healthy sample to those mentioned above were the presence of a schizophrenic disorder, an addictive or anxiety disorder, and the regular use of medication.

The final sample consisted of 20 healthy male control subjects (age: M = 25.25, SD = 3.23 years; school education: M = 12.55, SD = 1.61 years) and 20 schizophrenic patients (age: M = 25.25, SD = 3.23 years; school education: M = 12.55, SD = 1.61 years). The two groups did not differ significantly with respect to age and schooling (see Table 1). We controlled for current partnership and current fatherhood.

### 2.2. Measures

#### 2.2.1. Attachment Measure as the Basis for the fMRI Paradigm

Participants were administered with the fMRI-adapted version of the AAP [94] (test-retest reliability: Pearson’s r = 0.70, *p* < 0.001); 84% of the retest sample was classified in the same main categories). The AAP is a well-validated measure [32,95,96,97,98] that assesses adult attachment mental representation. This measure was used in two ways. The first was to determine participants’ attachment classifications and use the narratives to develop individualized individual narratives. The second was to use this material in the experimental fMRI paradigm (see below). The AAP is based on the analysis of “story” responses to a set of theoretically derived attachment-related drawings of scenes depicting solitude, illness, separation, death, and potential maltreatment. Drawings portray adults and children alone (three monadic pictures, representing abandonment) as well as adult-adult/adult-child dyads (four dyadic pictures, representing interpersonal distress) and one neutral picture. Individuals are asked to tell a story about each picture following a standardized set of interview probes. The classification is derived by evaluating the response patterns for the whole set of seven picture stimuli, each response of which is evaluated for content, discourse, and defensive processes. Organized attachment is defined in the AAP, following the attachment literature at large, as secure, and insecure-dismissing and preoccupied classifications; any frightening or threatening material that appears in the story is contained (e.g., desperately alone, death, attack, abuse)—note that not all stories contain this material. Transcripts are judged unresolved when there is no evidence of representational containment of frightening elements in at least one story. The nuances of coding are beyond the scope of this paper, and the reader is referred to George and West [36] for details. AAPs are transcribed verbatim from audio recordings for analysis.

The AAP has demonstrated solid psychometric properties, including test-retest reliability, inter-judge reliability, and convergent and discriminant validity [32,96,97,99].

#### 2.2.2. Clinical Measures

The Beck Depression Inventory-II (BDI, [100,101], internal consistency, α = 0.9), the State-Trait Anxiety Inventory (STAI, [102], internal consistency, α ≥ 86 ≤ 0.95), and the Positive and Negative Affect Schedule (PANAS, [103], internal consistency, α ≥ 0.84) were administered before scanning to assess depression, anxiety and mood in both groups. Between-group differences were examined by an ANOVA. Patients and healthy control participants showed no significant differences in state or trait anxiety as well as in negative mood. There were significant differences in depression and positive mood, whereupon patients had higher BDI-II and lower PANAS-PA scores. Although the difference in BDI-scores was highly significant between both groups, a mean BDI-score of 8.6 (SD = 6.3) in patients indicated that there were no clinically relevant depressive levels in the patient group (Cut-Off ≤ 10, [104]; Cut-Off ≤ 13; [101]).

We used a small neuropsychological battery to test patients and controls for handedness, attention, verbal IQ, and working memory capacity as differences in those cognitive functions could possibly interfere with performance in our task. Premorbid verbal IQ was assessed using the WST [104], internal consistency, α = 0.94. Handedness was determined by a questionnaire designed by Chapman and Chapman [105]. One subtest of the TAP (Testbatterie zur Aufmerksamkeitsprüfung, [106]; internal consistency, α = 0.70) “Alertness” was administered to measure basal attention as well as one subtest of the WAIS-III (Wechsler Adult Intelligence Scale: German version by von Aster et al. [107], internal consistency, α ≥ 0.73 ≤ 0.94), “Letter-Number-Span” to assess working memory performance. When conducting an exact Chi^2^-Test for handedness and an ANOVA for all other neuropsychological measurements no between-group differences in cognitive functioning could be found. Results are partly included in Table 1.

After a complete description of the study to the subjects, written informed consent was obtained. The protocol according to the Helsinki Declaration was approved by the local institutional ethics committee.

#### 2.2.3. Oxytocin Measurement

Blood samples were collected between 8 a.m. and 9 a.m. two to four weeks before fMRI measurement using EDTA tubes containing aprotinin 400 IU/ml blood to avoid hormone degradation. Samples were kept on ice for a maximum of 1 h until centrifugation at 4 °C at 1500× *g* for 15 min. Supernatants were collected and stored at −80 °C until being assayed (maximum 6 weeks). Oxytocin was measured using a commercially available immunoassay ELISA kit (Assay Design, Ann Arbor, MI, USA) according to the instructions of the supplier. Briefly, the samples were five-fold diluted in order to avoid matrix effects, 100 µL was used in the assay. Analyses were carried out in triplicates. The intra-assay coefficient of variation (CV) was 14.9% and the inter-assay CV, determined across 10 separate runs was 18.95%.

#### 2.2.4. Experimental Design

We used an event-related design (see Figure 1) for the study adapted from Buchheim and colleagues [94]. Stimuli were derived from the Adult Attachment Projective Picture System (AAP, [32,95]), an established and validated interview to assess attachment patterns in adults. The AAP consists of seven drawn picture stimuli, designed to activate the attachment system [32]. Two to four weeks prior to our fMRI experiment, AAP interviews were conducted by a trained investigator. The administration included asking participants in a semi-structured format to describe the scene in the picture, including what could have happened to cause the scene, what characters are thinking or feeling, and what they think might happen next in the scene. Three “core sentences”, that represented the attachment pattern of the participants were extracted from the audiotaped responses to each AAP picture stimulus by two independent certified judges (e.g., “A girl is incarcerated in that big room”, “A child with separated parents, one is leaving now”). These sentences were paired with the respective pictures to constitute the “personally relevant” trials tailored to each participant. The same pictures were paired to neutral, self-irrelevant sentences matched for length and describing only environmental aspects of the depicted situation (e.g. “There is a window with curtains on the left and right”, “in front of the window there is a house with a chimney”) and constituted the “neutral” trials, which were identical for all participants. Each trial consisted of the presentation of either the relevant or neutral sentence for 4.5 s, the corresponding picture for 4.9 s followed by a fixation cross for 9.9 s (see Figure 1). Participants were instructed to mentally engage with the attachment scene in the picture and its textual description. To ensure that the participants stayed focused on the task they were told to press a button with their right index finger after reading the sentence. Stimuli were presented within two blocks of six trials each. Each block contained three different sets of personally relevant sentences and neutral sentences for each picture stimulus. Trials alternated between personally relevant and neutral in groups of seven AAP picture stimuli. Pictures were presented in the same order as in the interview to activate the attachment system. In total, there were 84 trials resulting in a scanning time of about 28 min. After scanning, participants were asked by questionnaire to rate on a seven-point Likert scale the degree of emotional arousal and autobiographical relevance of personally related and neutral sentences. There were no significant differences in the ratings of personal relevant sentences between patients and controls, except patients rated neutral sentences as more emotionally arousing than controls (F [1,38] = 4.58, *p* < 0.039) (see also Haralanova et al. [108]).

#### 2.2.5. Data Aquisition

MRI data were obtained using a Philipps Achieva 3.0 T TX (Philips Medical Systems, Best, The Netherlands) scanner, equipped with a standard sense-8 head coil. High-resolution T1*-weighted 3D fast-field echo (FFE) sequences were obtained for anatomical reference (160 slices, TR/TE shortest, FOV 230 mm × 230 mm × 115.5 mm, matrix 240 × 240, voxel size 1 mm × 1 mm × 1 mm, flip angle 8°, sagittal acquisition order, duration 5 min). A total of 880 Echo Planar Imaging (EPI) T2*-weighted whole brain volumes were acquired (33 slices, TR/TE = 2000/35 ms, flip angle 90°, FOV 230 mm × 230 mm × 115.5 mm, matrix 76 × 77, voxel size 3 × 3 mm, slice thickness 3.5 mm, ascend sequential acquisition order, standard AC-PC orientation) split in 2 functional runs of 440 volumes each.

#### 2.2.6. Statistical Data Analysis

All statistical analyses of behavioral data were done using SPSS 21 (SPSS Inc., Chicago, IL, USA). Statistical analyses of fMRI data were carried out with SPM8 (www.fil.ion.ucl.ac.uk, released April 2009, accessed 1 May 2009) and MATLAB R2012a (MathWorks, Natick, MA, USA). Functional volumes were realigned to the first volume. After realignment, a mean EPI image was created, which was together with the functional data co-registered to the corresponding structural T1 image as well as a T1-template of standard stereotactic space defined by Montreal Neurological Institute (MNI) provided with SPM8, to facilitate normalization. Subsequently, images were spatially normalized to standard stereotactic space defined by MNI. Functional images were then smoothed with a 3D isotropic 8-mm full-width/half-maximum (FWHM) Gaussian kernel. Low-frequency noise was removed by applying a high-pass filter (cut-off 128 s) to the fMRI time series at each voxel. Statistical analysis was carried out using the general linear model with a delayed boxcar waveform to model blood oxygenation level-dependent (BOLD) signal changes of personal relevant trials to neutral trials for each individual. Thereby only pictures following relevant or neutral sentences were included as separate regressors in the model. Sentences were left out since we expected different kinds of neural responses to textual and imagery stimuli. Motion-correction parameters as well as ratings of emotional arousal of the relevant and neutral sentences were included in the model as parameters at the first level. For every subject, two contrasts were calculated: “pic_ir” (picture after irrelevant sentence > baseline); “pic_re” (picture after relevant sentence > baseline). Statistical parametric maps for each contrast were calculated on a voxel-by-voxel basis for each subject. Goodness-of-fit (beta) values for each contrast resulted in a contrast map for each individual. The statistical parametric maps from each individual data set were entered into second-level, random effects analyse accounting for subject-to-subject variability using a multi-subject repeated measures ANOVA “flexible factorial” 2 × 2 design (group vs. task) with 1 main effect and 1 interaction: main effect “task” (pic_ir/pic_re) and interaction (group/task). Additionally, BDI and PANAS positive affect scores were included as parameters to account for confounding intersubject variability of no interest. A voxel was deemed significant if its z-score was greater than 3.21 corresponding to *p* < 0.05 FDR-corrected. Additionally, cluster correction < 0.05 FDR was applied.

For correlation analysis of MRI data and behavioral as well as neuroendocrinological data, regions of interest (ROI) analysis was carried out focusing on the regions identified by whole-brain analysis. Statistical maps were limited to grey matter areas derived from Wake Forest University PickAtlas (www.fmri.wfubmc.edu, accessed 1 January 2008). The mean activation for every ROI of every subject was calculated. The resulting contrast values as measures of effect size were then correlated with oxytocin- and PANSS-scores as well as the duration of illness since 1st SPA using SPSS 21 (SPSS Inc.).

## 3. Results

### 3.1. Attachment Classifications

The coding of the attachment interviews [32] by a certified reliable judge (AB) revealed significant between-group differences in the distribution of the four attachment patterns measured by the AAP (secure, dismissing, preoccupied, unresolved) using exact Chi2 statistic (χ^2^ = 19.55; *p* < 0.000, see Figure 2). As expected, the patient group enclosed no secure individuals, while 60% (*n* = 12) of the control group showed secure attachment representations (χ^2^ = 21.95; *p* < 0.000). Importantly, patients displayed in accordance with Lavin et al. [12] a significantly higher percentage of an insecure-preoccupied pattern of attachment compared to controls (χ^2^ = 6.53; *p* < 0.031). Differences concerning the distribution of dismissing (χ^2^ = 0.63; *p* < 0.695) and unresolved (χ^2^ = 1.31; *p* < 0.451) individuals did not reach significance, whereupon both attachment patterns occurred prevalent in the patient group.

### 3.2. Endocrinological Data

When comparing endocrinological data between both groups, the Levene-Test for homogeneity of variances revealed a significant result for both measurements of OXT (L_Oxy1_ = 4.82, *p* < 0.034; L_Oxy2_ = 5.73, *p* < 0.022; L_Oxymean_ = 5.35, *p* < 0.026). Therefore, non-parametric-rank-based Wilcoxon-Mann-Whitney-Tests were used to compare oxytocin between patients and controls. According to our hypothesis, healthy controls showed higher OXT scores compared to patients at both points of measurement. For results see Table 3.

### 3.3. Neuroimaging Data

First, we assessed between-group differences of patients and healthy controls: patients (>healthy controls) revealed significantly higher activation when watching AAP pictures following personally relevant sentences versus AAP pictures following neutral sentences in the anterior and posterior cingulate cortex, insula, precuneus, temporoparietal junction, left premotor cortex, right supplemental motor area and left claustrum (Table 4, Figure 3). Participants of the control group did not show any higher activation compared to the patients.

Second, we performed a conjunction analysis to reveal shared activations of patients and healthy controls in the above contrast: one significant cluster located within the posterior cingulate gyrus was detected (Table 5).

Third, to assess associations between imaging data and clinical data as well as endocrinological data we correlated contrast values of regions identified by whole brain analysis (cingulate gyrus, insula, precuneus, TPJ) with PANSS-Scores (positive, negative), duration of illness since first SPA in patients and with oxytocin-scores in both groups. Correlation analysis gave evidence for a significant negative association between PANSS positive scores and activation of the bilateral precunei (r_lP_ = −0.458, *p* < 0.042; r_rP_ = −0.468, *p* < 0.037; see Figure 2) as well with the response of the left TPJ (r_lTPJ_ = −0.450, *p* < 0.047). Another significant negative correlation was found between activation of the bilateral insulae and the duration of illness since 1st SPA (r_lI_ = −0.481, *p* < 0.032; r_rI_ = −0.562, *p* < 0.001, Figure 3). At last, there was a significant positive correlation between oxytocin scores and activation of bilateral precunei (r_lP_ = 0.501, *p* < 0.025; r_rP_ = 0.520, *p* < 0.019) and bilateral TPJ (r_lTPJ_ = 0.462, *p* < 0.040; r_rTPJ_ = 0.543, *p* < 0.013) in patients but not in healthy controls (see Figure 4).

Finally, we wanted to explore differences in neuronal activations of organized and disorganized attachment patterns in patients even though there was no significantly higher percentage of unresolved individuals in the patient group than in controls. Because of the reduced number of subjects in each group in the analysis, we lowered the significance level to 0.001 (peak level) and an additional voxel threshold of 50 contiguous voxels (see Figure 5).

Disorganized/unresolved patients showed significantly higher activation in one cluster located in the right TPJ (Table 6). Organized/resolved patients displayed no higher activation than disorganized ones.

## 4. Discussion

With regard to our hypotheses, a higher proportion of insecure attachment in schizophrenic patients compared to secure individuals could be confirmed. As expected OXT levels in SZP were lower than in HC. On a neural level, we found increasing response in patients within a social mentalizing network when confronted with personally relevant sentences (priming) before ambiguous attachment-relevant pictures in self-reflection networks (precuneus, TPJ, insula, frontal areas). In our study, the patients’ responses might be considered a mode of dysregulation when triggered by this kind of personalized material. In the patient group, we found positive correlations between OXT and three brain areas (bilateral dlPFC, precuneus, left ACC). OXT might modulate within this network.

### 4.1. Discussion on Oxytocin Level

As expected, we found a higher amount of insecure attachment in SZP, but lower OXT concentration in the same patient sample. Both findings are in line with prior descriptions of schizophrenia and research results. According to our hypothesis, SZP had a lower OXT concentration in blood plasma than the HC at both measurement time points. These results are consistent with those of Rubin and colleagues [63] and Keri and colleagues [80], who also found a lower OXT concentration in the blood plasma of SZP. They contrast with the findings of Walss-Bass and colleagues [78], who found higher levels of OXT in their study in a relatively large group of SZP. They found the highest oxytocin levels in patients with pronounced delusions. In our sample, however, the patients had relatively few positive symptoms, which could be at least partly responsible for the difference. With regard to our results, however, it must be noted that the oxytocin levels varied widely between individuals, especially in the control group. This fact, however, is in line with the literature [109,110] and might be the main reason for the unclear findings with regard to OXT in SZP.

We also tested whether there was a linear negative correlative relationship between the level of oxytocin and the expression of patients’ positive and negative symptomatology. These hypotheses could not be confirmed. Similar findings were also reported by Goldman and colleagues [79] and Rubin and colleagues [63] in male schizophrenic patients.

### 4.2. Discussion on Attachment Pattern in Relation to Oxytocin

As assumed, the number of insecure attachment representations in SZP was significantly higher than in HC who only showed secure attachment representations. This is in line with previous meta-analyses [12,28,70]. We could not confirm a predominance of avoidant/dismissing attachment in our study. However, we found a higher amount of insecure-preoccupied attachment patterns was higher than the other insecure-attachment groups. This is in line with the systematic review by Lavin et al. [12] using self-report measures (anxious attachment style as the equivalent of preoccupied attachment). Preoccupied attachment is defined by hyperactivation of the attachment system represented by vagueness, topics of conflict, and anger in the attachment narratives. Moreover, our results demonstrate a higher number of unresolved patterns compared to HC, confirming the results by Bucci et al. [39], though they did not reach significance.

Our findings are in line with Abu-Akel [111] and Abu-Akel and Bailey [112] who theorize that in schizophrenia there is a so-called hyper-theory-of-mind ([112], p. 735) or hypermentalization ([113], p. 2979).

Hypermentalization means that schizophrenic patients are not only able to imagine that other people have mental states such as thoughts and feelings that influence and guide their actions (mentalizing, [114]), moreover, they are convinced that others are furthermore aware of patients’ mental states such as thoughts or desires, or that others can influence their thoughts and feelings. The theory of hypermentalization contradicts the common view of a number of authors [115,116,117], who attribute the difficulties of schizophrenic patients in mentalizing tasks [118] to a reduced ability to mentalize. In light of Abu-Akel’s theory [111,112], it would be conceivable that hypermentalization in schizophrenia is also associated with hyperactivation of the attachment and social mentalizing system, as hypothesized for insecure-unresolved attachment. Individuals with insecure-unresolved attachment have often experienced in their past that they are unable to face threats alone and consequently have a pronounced fear of separation. Such an enhanced fear response to separations would be consistent with our finding that schizophrenic patients have lower OXT levels than predominantly securely attached healthy subjects, given the fear-attenuating effects of OXT. In parallel, Pierrehumbert and colleagues [119] found lower plasma OXT concentrations under stress in individuals with insecure-unresolved attachment. This was not evident for individuals with other attachment patterns.

If we follow De Dreus’ [120] assumption that OXT improves the ability to differentiate between self and others, it would be conceivable that OXT improves the ability for social cognition and mentalization by sharpening the ability to differentiate between one’s own self and others or between one’s own needs and thoughts and those that one assumes others have. Evidence for this consideration is provided by a study by Colonnello et al. [121]: They were able to show that intranasal OXT application improved the ability of healthy males to distinguish their own face from other faces after it had been altered by morphing. This has been already shown in psychosis [79].

Conversely, this implies that low oxytocin levels are associated with a poorer ability to discriminate between one’s own ideas and thoughts and those of others. Abu-Akel’s concept of hypermentalization in schizophrenia [111,112] corresponds to an extreme form of confusing one’s own and others’ ideas or inside and outside and would possibly be related to patients’ OXT levels as well as insecure-unresolved attachment styles.

### 4.3. Discussion on fMRI Data

#### 4.3.1. Stimuli-Based Discussion of Brain Activation

As we used paradigm-inducing attachment-relevant social scenes we expect alterations/differences in brain areas concerning neural networks of self-reflection and autobiographical memory and self-relevant decision making including the precuneus, insula, TPJ, lateral and medial prefrontal cortex, limbic system, anterior cingulate gyrus (cognitive decision if relevant form or not), posterior cingulate (emotional judgment). Compared to the HC, the SZP showed an altered neuronal activation pattern when their binding system was activated with autobiographically relevant stimuli (compared to neutral stimuli).

#### 4.3.2. Group Comparison

In accordance with our hypotheses, we found significantly stronger neuronal activation in part of the brain areas associated with autobiographical memory processes, such as the precuneus, the PCC, and the TPJ, when patients viewed the AAP images paired with core sentences from their AAP interviews compared to viewing the images paired with neutral sentences. Contrary to our hypothesis, we did not find enhanced activation in the hippocampal area, the amygdala, or the STS. Furthermore, as hypothesized, patients showed enhanced neuronal activation in areas that are important for emotion regulation processes, such as the ACC and peripheral areas of the dlPFC. Finally, in line with our hypothesis an increased neuronal activation in empathy-related areas such as the insula, but not in the fusiform gyrus or the dmPFC, could be detected in the patients.

These brain regions, which are more activated in the patient group, are part of a network that is referred to in the literature as the mentalizing theory of mind network (or mentalization network [113,122]. Other brain regions in this network are the OFC, the vmPFC, the dmPFC, the ILFC, the amygdala, the striatum, and the STS [113,122,123]. This network is of elemental importance when we think about ourselves in relation to significant others. Abu-Akel and Shamay-Tsoory [113] hypothesize that the first distinction of whether a stimulus should be processed as related to self or others occurs in the TPJ. According to their model self-related mental states are processed in the precuneus and the PCC, and other-related states are processed in the STS. Subsequently, the neuronal signals are transmitted via the striatum and the amygdala to the ACC. Depending on whether the ideas or representations of the stimulus are more cognitive or affective, further mental processing follows one of two pathways: Cognitive representations follow a dorsal processing path, and affective representations follow a ventral processing path. The final resulting decision and evaluation processes of cognitive representations take place in the dmPFC, those of affective representations in the vmPFC, and the OFC (see Figure 6).

Together with the pPFC, the dlPFC plays an important role in attention control and emotion regulation. Furthermore, together with the PCC, it seems to be significantly involved in the inhibition of action tendencies [124] as well as in the establishment of associative links between contents of long-term memory [125,126]. The insula plays an important role in the processing of sensory and emotional stimuli, being significantly involved in interoception, the awareness of bodily states ([127], p. 2). In the course of this, it is also relevant for judging the self-consciousness of actions, i.e., whether or not a sensory event is influenced by our own actions. In this way, it allows us to distinguish between inside and outside at a relatively basal cognitive level [127,128,129]. In addition, the insula plays a role in the perception of pain in oneself as well as in others [130,131]. The ventral ACC also plays an important role in self- and emotion-regulatory processes [132] and in relation to the perception and regulation of pain experience [133,134].

The ACC also has functional connections to the limbic system [135]. Furthermore, the ACC is involved in social decision-making processes. Its task is to judge whether a stimulus is a reliable predictor of certain events or whether a stimulus is actually followed by a certain expected event. Apps et al. [136] showed that when a stimulus did not prove to be a reliable predictor of an event related to the subject himself, ventral parts of the ACC were activated. If, on the other hand, a stimulus was not a good predictor for an event related to others, dorsal parts of the ACC showed increased neuronal activation. Thus, similar to the insula, the ACC is involved in judging self- and other-reference to stimuli, but at a higher, more complex level of processing. In a study by D’Argembeau and colleagues [137], the dmPFC, the ACC, and the precuneus were found to be active in appraising oneself versus another person in relation to complex social features such as sociability.

In accordance, Sajonz and colleagues [138] described that the precuneus, as well as the PCC, showed an enhanced neural response to self-referential versus other-referential stimuli. Cavanna and Trimble [139] hypothesize that the task of the precuneus is to relate to past experiences with other self-related biographic episodic memory contents. This assumption is supported by the study of Sajonz and colleagues [138], as the precuneus was activated during the retrieval of episodic memory content in addition to self-referential stimuli, whereas the PCC was not. Brewer and colleagues [140] discuss the importance of the PCC for arising the feeling “caught up” in something and being unable to mentally detach from it. Thus, in addition to processing self-referential stimuli, the PCC plays an important role in the context of addictive pressure [141], resting networks [142], and when we mentally digress and think about our past or future [143,144]. At a basal level, its activity is associated with a decrease in attention to external processes and poorer performance in attention tasks in which one is expected to respond to external demands [145]. The results of a study by Sowden and Catmur [146] suggest that the right TPJ also has a specific role in the context of social cognition: the switching between self- and object representations or the attribution of a stimulus as coming from within or from outside by activating one perspective and inhibiting the other. Towards the background of these literature findings, the results of the group comparison suggest first of all that in the SZP, compared to HC, there is an increased activation of self-reference processes or hypermentalization in the sense of an increased tendency to relate stimuli to oneself as soon as their attachment system is activated.

In the group of HC, on the other hand, only an increased activation in the area of the PCC was found when viewing the pictures combined with core sentences compared to pictures with neutral sentences. This could be a correlate for a mental digression into one’s own past and would be plausible due to the autobiographical relevance of the core sentences. However, HC showed also a number of deactivations in the above-mentioned mentalization network, predominantly left hemispheric in the area of TPJ and precuneus, striatum, and thalamus as well as in frontal and orbitofrontal areas. Based on Abu-Akel and Shamay-Tsoory’s model ([113], Figure 6), this can be interpreted to mean that there is reduced self-referencing or inhibition of areas significant for establishing self-reference in healthy individuals. This seems counterintuitive at first. Thus, one would assume that activation of the binding system in HC would lead to increased self-referencing, but maybe less effortful and less affect-laden as in SZP. In our study, however, it should be kept in mind that the AAP interview not only aims to activate the attachment system but also to put the interviewee under attachment stress, i.e., to evoke aversive feelings.

The insula and dmPFC are important areas known to be crucial for the development of emotion. In a study by Ochsner and colleagues [130] on the perception of pain, the subjects were either subjected to mild physical pain themselves in the form of thermal stimulation or were shown videos of accidents of other persons. In both conditions, activation of dmPFC, insula, thalamus, and PC occurred, with dmPFC and insula showing greater activation when the pain was self-inflicted on the subject. The inhibition of self-referencing processes could therefore represent a kind of regulatory strategy with the aim of regulating an intensified immersion in unpleasant autobiographical attachment episodes. In this way, the emergence of the unpleasant feelings associated with these episodes could be modulated. Neuronally, this could in turn translate into inhibition of areas associated with the perception of pain. Evidence for this interpretation is provided by a study by Nolte and colleagues [147]: Healthy subjects processed a revised version of the eye-particle test in the fMRI scanner. They had either to estimate the age of the presented persons on the basis of their eye parts or to judge their mental state or affect according to the original test. Prior to this, either attachment-related or non-attachment-related stress was induced in each subject using an appropriate tape script tailored for them. The authors found decreased activation in the left STS, left TPJ and left OFC in healthy subjects after the induction of attachment-related stress versus non-attachment-related stress. In parallel with the present study, this may suggest that attachment-related stress or its anticipation is associated with the inhibition of parts of the mentalization network.

In SZP when viewing APP images according to core sentences activation patterns of brain areas associated with the development of emotions and the experience of pain, such as dmPFC, amygdala, and insula [130,148,149] and PCC [150,151] were found. This could mean that there was an increased occurrence of negative emotions related to attachment-related autobiographical memory content in the patients. Furthermore, increased activation of the OFC and the STS was evident within the patient group. In Abu-Akel and Shamay-Tsoory’s [113] model, the OFC, the dmPFC, and the STS form a subnetwork for processing the mental states of other individuals. Taken together, this could mean that activation of the attachment system in schizophrenic patients may lead to increased emergence of negative emotions and difficulty in mentalizing other minds due to dysfunctional early attachment experiences.

On the behavioral level, this hyperactivation in the sense of Abu-Akel and Shamay-Tsoory’s [113] concept of hypermentalization could lead to temporary impairments in the ability to correctly distinguish between internal and external stimuli or to a tendency to mistakenly interpret internal, self-referential stimuli as external, other-referential ones. This would correspond to the positive symptoms of schizophrenic patients in the form of ego disorders or hallucinations.

The conclusion that activation of the attachment system is accompanied by an activation of the mentalization network seems plausible towards the background of attachment theory and AAP-based stimuli. The latter are designed in such a way that the viewer can easily identify with the depicted figures and in this way, memory processes of attachment episodes from the own past are stimulated. As already shown, other patient groups, such as depressive patients [94] or patients with borderline personality disorder [152] also show increased neuronal activation in parts of their attachment network compared to HC. Inconsistent results due to different paradigms.

#### 4.3.3. OXT-Based Discussion of Brain Activation

The assumption of a significant negative or positive correlative relationship between brain activation and oxytocin levels could not be confirmed. However, a trend could be observed that the higher the oxytocin levels of the patients, the higher the activation in their bilateral TPJ as well as their left precuneus. In Abu-Akel and Shamay-Tsoory’s [113] model, the TPJ takes is a mediator between self- and other-referential processing pathways: it has structural connections to the thalamus, the limbic system, and visual, auditory, and interoceptive brain regions, likely integrating information from these areas and relaying it to temporal and prefrontal cortex areas. Accordingly, the TPJ forms a hub where both signals from the external world and interoceptive percepts are integrated and play a central role in the classification of stimuli into self-referential and other-referential. Evidence for these considerations is provided by a meta-analysis by van Veluw and Chance [129]. The authors demonstrated that the TPJ is activated both in tasks involving the processing of stimuli that refer to other people and in tasks involving the processing of self-referential stimuli. There is evidence that oxytocin intake in competitive situations reinforces the tendency to classify others according to whether they belong to one’s own group or to a foreign, competitive group. Goh and Lu [122] considered oxytocin as a potential predictor of ToM and social functioning in patients with schizophrenia.

#### 4.3.4. Attachment-Based Discussion of Brain Activation

An increased presence of insecure attachment styles is usually associated with unpleasant relationship experiences during childhood and adolescence. Our findings, hence, could indicate a connection between the increased activity of the mentalization network and negative relationship experiences in the past.

Our main finding of increased TPJ activation in SZP with unresolved attachment patterns is consistent with previous findings in healthy subjects that the attachment style is predicting the involvement of TPJ [153]. Baskak and colleagues showed that developed attachment styles do have an effect on the representation of ToM in terms of cortical activity in late adolescence. In their study avoidant/dismissive attachment is represented by lower activity in the right superior temporal cortex during ToM, which might be related to weaker social need and habitual unwillingness for closeness among their investigated group of adolescents. In contrast to Baskak and colleagues, we used attachment-related individual stimuli and probably stimulated the attachment-related memory network of aversive self-related critical life incidents that is reflected by an increased TPJ response in SZP compared to healthy controls. Though SZPs normally make an effort to protect themselves by avoiding stimulation of adverse memories and overwhelming self-related emotions, former studies on neutral stimuli showed that SZPs are even responsive to neutral stimuli by precepting them as aversive [108].

#### 4.3.5. Psychosis-Based Discussion of Brain Activation

The altered neuronal activation pattern showed a significant correlative relationship to the patients’ positive symptoms. A negative correlation was detected between the bilateral insula activation of the patients and their disease duration. This means that insula activation decreased with increasing disease duration. There is evidence in the literature for decreased volumes of the insular region of schizophrenic patients These differences in volume exist prior to illness onset and continue to increase as the illness progresses [154,155]. In addition, there is evidence of decreased cortical density in the insular region of schizophrenic patients [156,157,158]. In their review, Wylie and Tregellas [127] discuss the hypothesis that deficits in recognizing emotions in faces, in pain perception, and in the attribution of sensory percepts are related to dysfunctions in the insula in SZP. Recently, several authors [158,159,160,161] reviewed the relationship between insular and cognitive impairment and interoception in SZP. In light of these literature findings, it is conceivable that activation of the binding system due to pre-existing dysfunctions in the insula leads to hyperactivation of the insula, as is also found in auditory hallucinations [162]. The compensatory hyperactivation cannot be down-regulated and contributes, among other things, to an impairment of self-non-self-differentiation on a sensory level [163]. With increasing disease duration, there may be a decrease in this hyperactivation as we observed due to progressive structural changes in the insular region. Furthermore, in parallel to our hypotheses, a significant correlation between brain activation and positive symptomatology was found in the patient group when looking at pictures according to core sentences: The higher the activation in the bilateral precuneus was pronounced, the lower the positive symptomatology proved to be. In the Abu-Akel and Shamay-Tsoorys [113] model, the network of precuneus, PCC, IPL, and vmPFC serves predominantly to process self-referential stimuli. In light of our prior considerations that schizophrenic patients respond to activation of their attachment system with increased activation of other-referential areas, it seems plausible that such patients who also responded to attachment stress with activation of self-referential areas such as the precuneus had a lower tendency to confuse self- and other-reference or suffered from less positive symptomatology.

In connection with considerations on the origin of the self as well as the results of no systematic correlation between the thought disorders and the positive symptoms in SZP, this can be interpreted as an indication of an attachment-related, intersubjectively related to the origin of the disturbance of self-experience in schizophrenia.

#### 4.3.6. General Discussion

According to contemporary conceptualizations, psychosis is a neurodevelopmental disorder emerging during late adolescence and/or early adulthood and associated with the final stages of brain maturation [29]. However, neuroscience research suggests on the one hand that experiences of social adversity during early childhood, such as attachment-related adverse experiences (alone during threatening situations) or trauma [164], may also contribute to alterations of neural development and brain dysmaturational processes during adolescence/early adulthood in those who go on to have psychosis [65]. On the other hand, Brent et al. [165], Brent and Fonagy [166], and Debbané et al. [29] provided support for future research regarding two inter-related, early putative protective factors of developing a psychosis: attachment security and mentalizing (a social cognitive capacity fostered by attachment security), which may together heighten resilience to developmental interpersonal stress and moderate the risk for psychosis onset.

### 4.4. Limitations

Our sample is characterized by some specific patterns:(a)With regard to psychotic symptomatology: The mean PANSS total score was about 60 points, therefore the patient sample can be considered psychopathological stable. According to Leucht and colleagues [93], scores between 43 and 61 correspond to a clinical global impression (CGI, [167]) of three or mildly ill ([93], p. 234). The highest value corresponded to a CGI of four, i.e., moderately ill ([93], p. 235), and the lowest to a CGI of two, respectively on the borderline of mental illness ([93], p. 234).(b)With regard to medication: The intake of atypical antipsychotic drugs might have affected OXT plasma levels. With the exception of one patient, all patients were medicated, so that, as expected, there was little positive symptomatology such as hallucinations and delusions, but a certain degree of negative symptomatology such as social and emotional withdrawal and flattening of affect. A point of criticism to be discussed in this context is the fact that one patient was not receiving neuroleptic medication at the time of data collection. This is a potential confounding variable. However, the non-medicated patient in our sample did not show any abnormal results compared to the other medicated patients with respect to his neuropsychological functioning level or the pattern of his brain activation in the two fMRI paradigms.(c)Sample characteristics: Our findings are limited by sample size, since in both groups only 20 subjects could be finally included for statistical analysis. Further studies have to be conducted to provide more evidence in larger samples and for differentiating between different subtypes of psychosis and different states of disease and separating by considering the endurance of the disease since the first episode. Our results might be gender-related since we included only male patients and cannot be transferred to female patients. The results therefore cannot be generalized to a general schizophrenic population. Nevertheless, our findings indicate that an unresolved attachment style in schizophrenia is related to less concentrated OXT and to a cluster of increased brain activation within the mentalizing neural network.(d)Since a cross-sectional design is used, behavioral and neurophysiological parameters cannot be analyzed over time. To conclude cause- and effect relationships is difficult because the data are based on a one-time measurement of both the alleged cause and effect.(e)With regard to attachment representations a limited generalizability can be assumed and has to be addressed in further studies. However, we selected one of the most widely accepted, well-validated methods of assessing individual differences in attachment representations in adolescence and adulthood.

## 5. Conclusions

### 5.1. Conclusion with Regard to Developmentals Aspects

Research on the social complexity of rearing environments in rats suggests that environments rich in social and cognitive complexity are associated with significantly more synapses per neuron throughout the visual cortex compared to simple socially paired housing and individual housing [168]. These effects remained even after later environments were changed or reversed, suggesting that plastic changes associated with early experiences are persistent. Throughout the earliest stages of attachment, at least two brain structures, the locus coeruleus, and the amygdala, interact to facilitate the familiarity and reinforcement associated with the caregiver in filial bonding.

Parents and their infants are physiologically entrained, with patterns of heartbeat, hypothalamic–pituitary–adrenal activation, and oxytocin levels mirroring and responding to each other. Security-providing parents intuitively know and are rewarded by their infants, skillfully regulating their needs and affects. For insecure caregivers, negative effects in their own children and others evoke rejection and diminished oxytocin reward. Clinical and animal models [169] show that attachment trauma and other adverse experiences in early life influence the developing brain with long-term psychopathological consequences.

### 5.2. Conclusions with Regard to Prevention and Psychotherapy

Studies (see also Holmes and Slade [1]) of the physiology of therapist–client attachment are rare, but it is likely that a form of biobehavioral synchronization also plays a significant role in the success of psychotherapy, especially at mutative moments in sessions. Therapists vary widely in their effectiveness [170]: therapists can, due to their attachment, either react intrusively with overempathic responses or make cuttingly dismissive interpretations, however seemingly accurate [171]. As a meta-model, attachment-informed psychotherapy ([1,172], AIP) emphasizes the centrality, specificity, and continuity of relationships and the modulation of threat. Attachment skills of therapists include the ability to recognize that threat triggers attachment behavior and is incompatible with emotional exploration and the ability to co-regulate affective states of hyper- or hypo-arousal [173]. Psychotherapy typically helps people at times of transition as a preventive intervention. Secure attachment makes the uncertainties of novelty tolerable, and resolvable with the help of an attachment figure. In insecure attachment, agency is compromised, while rigid prior probabilities are clung to, either by isolationism or over-dependency. In a long-term therapeutic relationship, as therapists’ and clients’ oxytocin systems become entrained, this cross-talks to the dopamine system. This could be preventive in high-risk-populations or even in schizophrenic patients to regulate affective and arousing states in interpersonal relationships.

### 5.3. With Regard to a Key Contemporary Attachment Concept: Mentalising

Mentalizing as an attachment-related concept focuses on prefrontal cortex–amygdala connectivity. In educational and psychotherapeutic contexts, arousal is managed with the help of a trusted intimate other or secure base, enabling the individual to think about thoughts and feelings, one’s own and those of others, and the interactions between them [174]. Hence, two brains are better than one for working on interaction and binding. In SZP, however, deficits in explicit and implicit mentalizing are described by Csulak et al. [175]. Based on the research results, systematic reviews, and meta-analyses, the intention attribution of the patients is damaged [176,177,178]. Mentalizing impairments are characteristic both in the acute and remission phases, and they can be detected in first-degree, clinically asymptomatic relatives [46,118,179,180]. Mentalizing may be deficient even before the onset of the disease, may predict psychotic conversion, and often worsens before the first episode [118,181,182]. Long-term studies of social functionality also suggest that functionality is already weaker in childhood and deteriorates markedly further in adolescence, which in turn significantly predicts impaired functionality over a 20-year period [183]. It may have even a therapeutic significance [184], as unaffected implicit mentalizing skills may represent a significant base for remediating the impaired explicit mentalizing skills. However, impaired implicit mentalizing can be a significant limit in remediation. Recently, Langdon et al. [184] highlighted the therapeutic implication of the differential effects of implicit and explicit aspects of mentalizing, as the remediation of explicit mentalizing may require interventions to strengthen compensatory strategies, while implicit mentalizing may require a more basic approach using techniques to improve attentional processes to support a more efficacious detection of agency signals. This could be true in SZP.

Even if it is important to note however that more studies are needed for a better understanding of the relationship between oxytocin levels, mentalizing and attachment patterns and psychotic symptomatology, the contribution of this study is that it provides important insight into the relationship between clinical, behavioral and neurophysiological patterns. If we can assume that individuals do, in fact, develop relationships with others in a way consistent with generalized representations of biographical experiences based on mentalizing capabilities, a guide for prevention and interventions could be developed. This might have an impact on the development and maintenance of psychotic symptoms already during adolescence.

## Figures and Tables

**Figure 1 brainsci-13-01125-f001:**
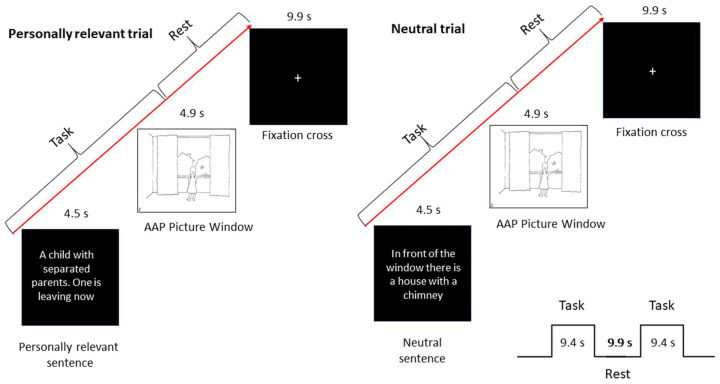
Stimuli were presented within two blocks of six trials each. Each block contained three different sets of personally relevant sentences and neutral sentences for each picture stimulus. Trials alternated between personally relevant and neutral in groups of seven AAP picture stimuli. Pictures were presented in the same order as in the interview to activate the attachment system. In total, there were 84 trials resulting in a scanning time of about 28 min. One measurement consists of 2 blocks of 6 single trials each with 440 scans for each functional run. After each trial started the sentence and the picture were presented for a total of 9.4 s. This was followed by 9.9 s of fixation (inter-stimulus interval).

**Figure 2 brainsci-13-01125-f002:**
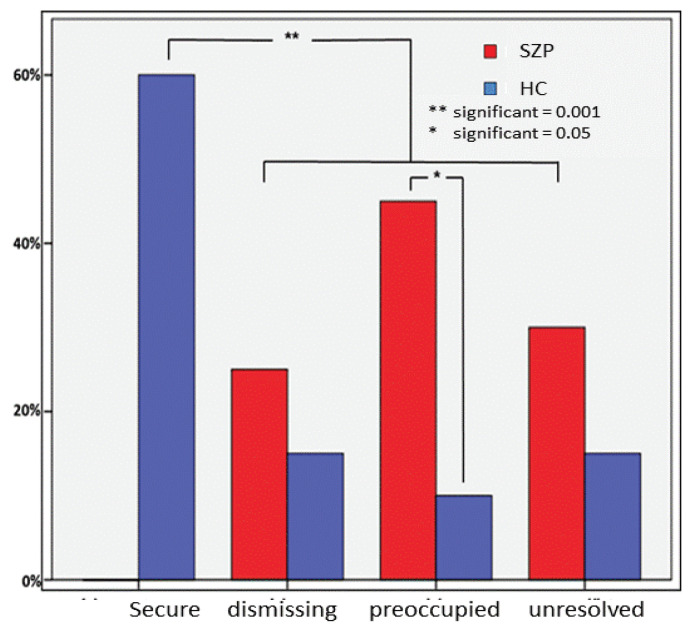
Proportional distribution of attachment classifications of patients and healthy controls by χ^2^-testing.

**Figure 3 brainsci-13-01125-f003:**
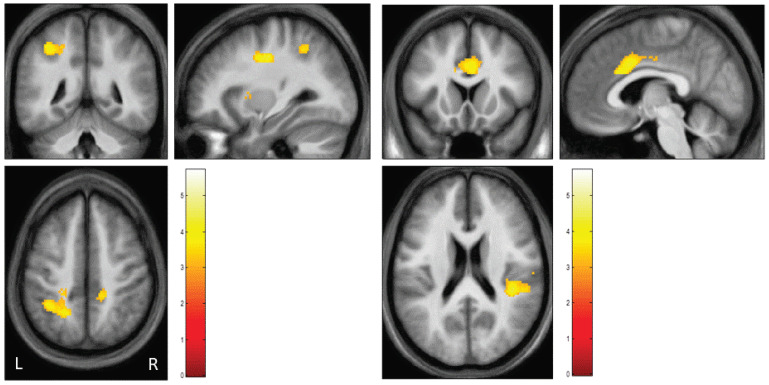
T-contrast personally relevant > neutral, activations SZP > HC: bilateral precuneus and temporoparietal junction; cingulate gyrus and insula (Peak-level: *p* < 0.05, FDR-corrected, cluster level: *p* < 0.05, FDR-corrected; L = left, R = right; statistical pictures showing the activation on a color scale from dark red to light yellow = statistical with yellow symbolizing stronger activation and red symbolizing weaker activation).

**Figure 4 brainsci-13-01125-f004:**
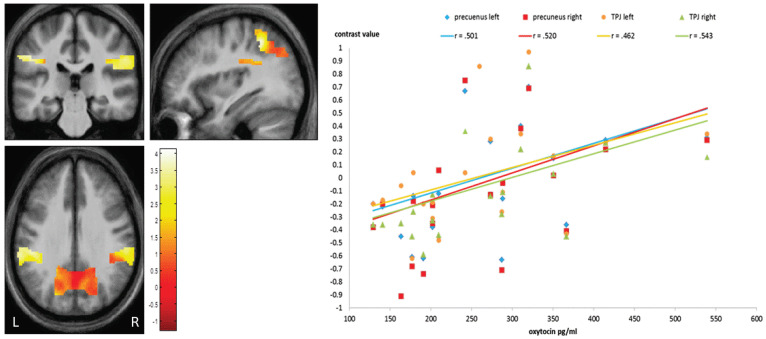
Positive correlation between oxytocin scores and ROI activation of (r_lP_ = 0.501, *p* < 0.025; r_rP_ = 0.520, *p* < 0.019) and bilateral TPJ (r_lTPJ_ = 0.462, *p* < 0.040; r_rTPJ_ = 0.543, *p* < 0.013; L = left, R = right) in patients, statistical pictures showing the activation on a color scale from dark red to light yellow = statistical with yellow symbolizing stronger activation and red symbolizing weaker activation.

**Figure 5 brainsci-13-01125-f005:**
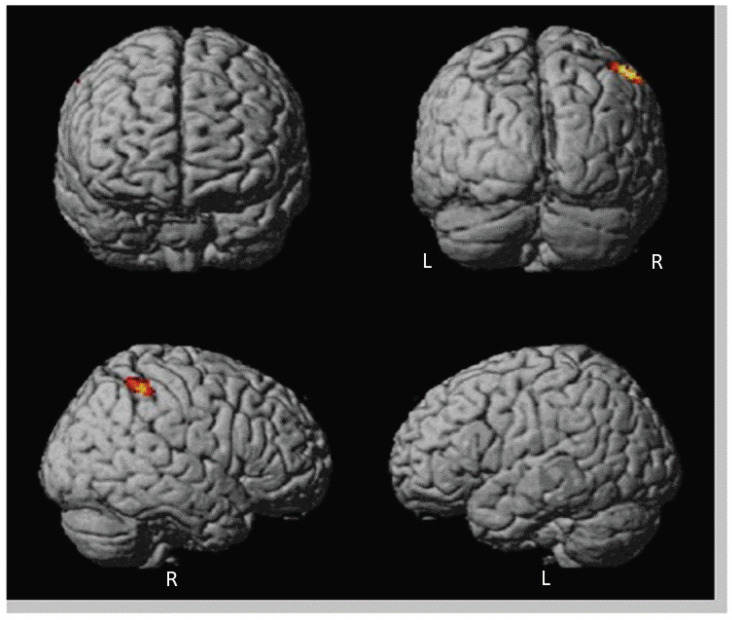
T-contrast personally relevant > neutral, activations disorganized/unresolved patients > organized/resolved patients (Peak-level: *p* < 0.001, uncorrected, additional threshold: 50 voxels, R = right, L = left; statistical pictures showing the activation on a color scale from dark red to light yellow = statistical with yellow symbolizing stronger activation and red symbolizing weaker activation).

**Figure 6 brainsci-13-01125-f006:**
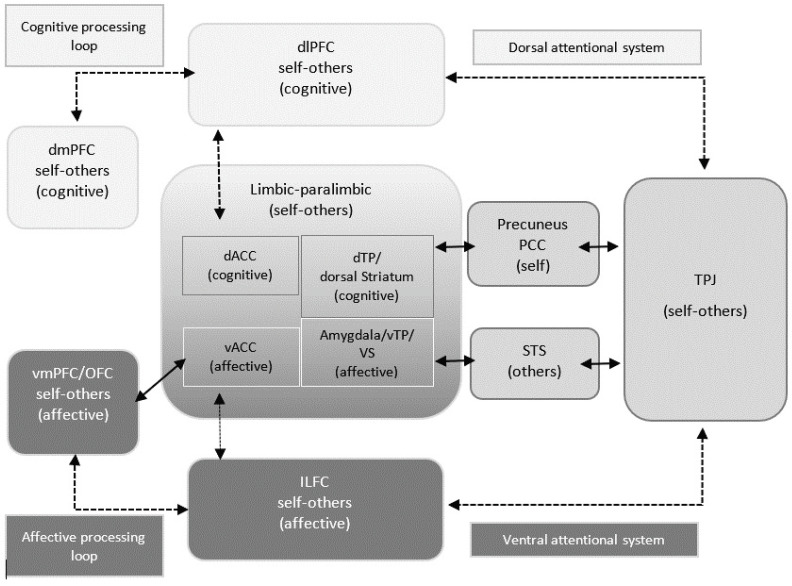
ToM-Network of mentalization (adapted from Abu-Akel and Shamay-Tsoory [113]).

**Table 1 brainsci-13-01125-t001:** Description of the sample.

	Patients (N = 20)	Controls (N = 20)	Statistics	Significance
Age (Mean, SD)	24.3 (3.7)	25.3 (3.2)	0.8 ^2^	0.370 (n.s.)
	(range: 18–30)	(range: 19–30)		
Years of education	12.2 (1.5)	12.6 (1.6)	0.5 ^2^	0.482 (n.s.)
Handedness	right:17	right:17	2.57 ^3^	0.349 (n.s.)
	left: 1	left: 3		
	ambydexter: 2	ambydexter: 0		
VIQ (WST)	106.3 (9.8)	111.3 (11.7)	2.2 ^2^	0.150 (n.s.)
STAI-T	40.7 (11.2) ^1^	36.5 (8.2) ^1^	1.8 ^2^	0.189 (n.s.)
STAI-S	39.4 (7.5) ^1^	40.1 (8.3) ^1^	0.1 ^2^	0.781 (n.s.)
PANAS				
PA	30.2 (6.3) ^1^	34.9 (4.8) ^1^	7.0 ^2^	0.012 (*)
NA	19.6 (4.8) ^1^	18.8 (6.7) ^1^	0.2 ^2^	0.666 (n.s.)

Note: all values are mean values and standard deviations in paratheses, no differences were found in working memory and alertness. ^1^ raw scores, not standardized, ^2^ median or f-value with df_1_ = 1 and df_2_ = 38, ^3^ exact χ^2^ value, * significant between-group difference for α = 0.05

**Table 2 brainsci-13-01125-t002:** Patient Diagnosis Classification.

		Statistics (Between Diagnosis Group Difference)	Significance
Diagnosis/	paranoid: 13×		
schizophrenia	disorganized: 5×		
subtype	undifferentiated: 2×		
Duration of illness since 1. SPA	3.96 (3.18)	0.4 ^1^	0.701 (n. s.)
PANSS total positive negative	55.9 (13.4) 9.75 (2.5) 18.6 (5.5)	3.95 ^2^ 3.84 ^2^ 2.19 ^2^	0.139 (n. s.) 0.147 (n. s.) 0.334 (n. s.)

^1^ f-value with df_1_ = 2 and df_2_ = 1 × 7, ^2^ χ^2^-value.

**Table 3 brainsci-13-01125-t003:** Comparison of endocrinological data of both groups.

	Patients (N = 20)	Controls (N = 20)	Statistics	Significance
Oxytocin (1st measurement)	266.5 (106.7) pg/mL	386.6 (221.4) pg/mL	−2.0 ^1^	0.043 (*)
Oxytocin (2nd measurement)	258.1 (105.6) pg/mL	385.8 (200.4) pg/mL	−2.2 ^1^	0.028 (*)
Oxytocin (mean)	262.3 (102.0) pg/mL	386.2 (208.3) pg/mL	−2.1 ^1^	0.033 (*)

^1^ z-value, * significant between-group difference for α = 0.05.

**Table 4 brainsci-13-01125-t004:** Significant peak activations (Peak-level: *p* < 0.05, FDR-corrected, cluster level: *p* < 0.05, FDR-corrected) SZP > HC: personally relevant > neutral, pictures only.

	Brodmann Area	Hemisphere	Z-Scores	Cluster-Size	MNI-Coordinates (x,y,z)		
Cingulate gyrus	BA24 BA31 BA32	L	4.56	2129	−16	−10	38
		R	4.10		6	10	34
		L	5.18		−16	−18	38
		R	3.80		18	−36	44
		L	3.30		−10	20	32
		R	3.85		4	22	28
Precuneus	BA7	L	3.82	2129	−20	−52	48
		R	3.10		12	−38	54
TPJ	BA40	L	3.89	2129	−36	−46	48
		R	3.21	565	54	−30	28
Precentral gyrus	BA6	L	4.36	2129	−32	−10	38
	BA4	L	4.16		−30	−14	40
Postcentral gyrus	BA2	R	4.01	565	60	−18	26
Insula	BA13	L	3.69	402	−48	−16	14
		R	4.06	565	42	−32	20
Claustrum	-	L	4.57	2129	−38	−4	6

**Table 5 brainsci-13-01125-t005:** Significant peak activations (Peak-level: *p* < 0.05, FDR-corrected, Cluster level: *p* < 0.05, FDR-corrected) conjunction analysis patient and controls personally relevant > neutral, pictures only.

	Brodmann Area	Hemisphere	Z-Scores	Cluster-Size	MNI-Coordinates (x,y,z)		
Posterior cingulate gyrus	23/31	L	4.60	137	−6	−50	24

**Table 6 brainsci-13-01125-t006:** Significant peak activations (Peak-level: *p* < 0.001, uncorrected, cluster threshold: 5 voxels) unresolved > resolved patient group personally relevant > neutral, pictures only.

	Brodmann Area	Hemisphere	Z-Scores	Cluster-Size	MNI-Coordinates (x,y,z)		
TPJ	40	R	3.98	92	50	−48	56

## Data Availability

The data can be obtained from corresponding authors.

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
