# Peer review of "The Interplay of Oxytocin and Attachment in Schizophrenic Patients: An fMRI Study"

_brainsci, 2023, doi:10.3390/brainsci13081125_

Round 1

Reviewer 1 Report

1. The diagram of the experimental paradigm (Figure 1) should be clearer and more detailed. The relationship among block, trial, stimuli, and rest, and their time sequence could be clearly described in one figure.

2. I wonder why FDR and FWE correction were both used in this study. This is not a commonly used strategy.

3. Why the results with p<0.04 were reported?

4. ”FEW corrected“ in Line 438 should be cerrected to "FWE corrected".

5. Several figures should be improved. For example, the meaning of the colorbars in figure 3,4 should be clearly expressed; the hemisphere should be marked in figure 3,4,5.

Author Response

Thank you for your very helpful remarks on our manuscript. We addressed all of them as you can find within the manuscript and in our response letter.

  1. The diagram of the experimental paradigm (Figure 1) should be clearer and more detailed. The relationship among block, trial, stimuli, and rest, and their time sequence could be clearly described in one figure.

Response: Thank you for this suggestion. We added the relevant information in the figure description and added more information in the figure itself.

  1. I wonder why FDR and FWE correction were both used in this study. This is not a commonly used strategy.

Response: Thank you for this valuable comment. Whether one chooses to control the FWE or the FDR ultimately depends on the situation or simply personal preference. However, we first did both analyses to get an idea about our data. Finally, there was no combination of FDR and FWE in our analyses that are presented in the manuscript. It was a mistake in the description that was done at a prior level of data analyses but there was no combination of both statistical strategies in the final data analysis. All comparisons were done with p < .05. In our study we used an FDR-corrected voxelwise threshold of p < .05 threshold, which would indicate that the current results have a 5% chance of containing a false positive. This is much more informative than using an uncorrected threshold of p < .05, which does not give any information about the expected rate of false positives. While the described multiple comparison correction methods aim to control the family-wise error rate, the false discovery rate (FDR) approach (Benjamini & Hochberg, 1995) uses a different statistical logic, and has been proposed for fMRI analysis by Genovese and colleagues (2002). In this approach, not the overall number of false positive voxels is controlled but the number of false positive voxels among the subset of voxels labeled as significant. The intuition behind FDR is that the proportion of false positive results is known. So, using a voxelwise FWE-corrected threshold of p < .05 means that the probability of any voxel being a false positive is 5% (hence, it is unlikely you are reporting any false positives). Using a voxelwise FDR-corrected threshold of q < .05 means that you expect 5% of your reported significant results to be false positives (hence, it is likely you are reporting false positives, but you know how many you are reporting).

  1. Why the results with p<0.04 were reported?

Response: Thank you, this was a mistake, we corrected and wrote p<0.05

  1. ”FEW corrected“ in Line 438 should be corrected to "FWE corrected".

Response: Thank you, since we corrected the whole section and our analysis is based on false discovery rate (FDR), this correction wasn’t needed.

  1. Several figures should be improved. For example, the meaning of the colorbars in figure 3,4 should be clearly expressed; the hemisphere should be marked in figure 3,4,5.

Response: We agree and added the meaning of the colorbars in figure 3,4 and added the hemisphere information.

Reviewer 2 Report

This is an interesting fMRI study on the relation between oxytocin and attachment in schizophrenic patients. The paper is well-written and the study was carefully conducted. I am impressed with the comprehensiveness of the paper. I agree that it will contribute well to the literature. I only have a few comments to improve the manuscript further:

1. One main concern that I have about this study is the relatively small sample. However, I appreciate that the authors have acknowledged this in the limitation and making it the sample clear in the abstract. Other than the sample size issue, it will be useful for the authors to highlight other limitations such as the use of cross-sectional design and limited generalizability of attachment representations.

2. There should be more information on how the control group was recruited. More information on their characteristics will be useful beyond what has been reported in the Table 1

3. It will be useful to add error bars in Figure 2

4. The internal consistency of the measures (especially clinical measures) used in the current study should be reported.

5. The conclusion in the abstract seems to end awkwardly. The authors should also use correlation wording rather than causal wording in the conclusion.

Author Response

Thank you for your valuable and helpful remarks. We addressed all of them as you can find within the manuscript and in our letter of response.

This is an interesting fMRI study on the relation between oxytocin and attachment in schizophrenic patients. The paper is well-written and the study was carefully conducted. I am impressed with the comprehensiveness of the paper. I agree that it will contribute well to the literature. I only have a few comments to improve the manuscript further:

  1. One main concern that I have about this study is the relatively small sample. However, I appreciate that the authors have acknowledged this in the limitation and making it the sample clear in the abstract. Other than the sample size issue, it will be useful for the authors to highlight other limitations such as the use of cross-sectional design and limited generalizability of attachment representations.

Response:  We fully agree with the reviewer and mentioned the small sample in the abstract and added additional limitations at the end of in section 4.4.

  1. There should be more information on how the control group was recruited. More information on their characteristics will be useful beyond what has been reported in the Table 1

Response:  Thank you for this suggestion. We added more information on how the control group was recruited in section 2.1.

  1. It will be useful to add error bars in Figure 2

Response: Thank you, however since we present the proportional distributions, we assume that isn’t needed to present error bars.

  1. The internal consistency of the measures (especially clinical measures) used in the current study should be reported.

Response: We agree with the reviewer and added overall internal consistency values of the clinical measures

  1. The conclusion in the abstract seems to end awkwardly. The authors should also use correlation wording rather than causal wording in the conclusion.

Response: Thank you for this valuable comment. We added a more comprehensive sentence in the abstract and adapted wording in the conclusion section.

Round 2

Reviewer 2 Report

The authors have addressed well all my comments. I appreciate all their efforts.